# Efficacy of novel SARS-CoV-2 rapid antigen tests in the era of omicron outbreak

**Kristin Widyasari[1], Sunjoo Kim**[1,2,3]*

**1** Gyeongsang Institute of Medical Science, Gyeongsang National University, Jinju, Republic of Korea, **2** Department of Laboratory Medicine, College of Medicine, Gyeongsang National University, Jinju, Republic of Korea, **3** Department of Laboratory Medicine, Gyeongsang National University Changwon Hospital, Changwon, Republic of Korea

* sjkim8239@hanmail.net

**Data Availability Statement:** All relevant data are within the paper.

**Funding:** This research was supported by the National Research Foundation (NRF) of Korea (NRF-2021R1I1A3044483 and NRF-2021M3E5E3080382). The funders had no role in

## Abstract

Following the outbreak of Omicron and its subvariants, many of the currently available rapid Ag tests (RATs) showed a decrease in clinical performance. In this study, we evaluated the clinical sensitivity of the SARS-CoV-2 Rapid Antigen Test 2.0 for nasopharyngeal swabs and SARS-CoV-2 Rapid Antigen Test 2.0 Nasal for nasal swabs in 56 symptomatic individuals by comparing the results between RATs, RT-PCR, Omicron RT-PCR, and whole-genome sequencing (WGS). Furthermore, sequences of the Omicron subvariants' spike proteins were subjected to phylogenetic analysis. Both novel RATs demonstrated a high sensitivity of up to 92.86%, (95% CI 82.71%– 98.02%), 94.23%, (95% CI 83.07%– 98.49%), and 97.95% (95% CI 87.76%– 99.89%) compared to the RT-PCR, Omicron RT-PCR, and WGS, respectively. The clinical sensitivity of RATs was at its highest when the Ct value was restricted to 15≤Ct<25, with a sensitivity of 97.05% for *RdRp* genes. The Omicron RT-PCR analysis revealed subvariants BA.4 or BA.5 (76.8%) and BA.2.75 (16.1%). Subsequently, the WGS analysis identified BA.5 (65.5%) as the dominant subvariant. Phylogenetic analysis of the spike protein of Omicron's subvariants showed a close relationship between BA.4, BA.5, and BA.2.75. These results demonstrated that SARS-CoV-2 Rapid Antigen Test 2.0 and SARS-CoV-2 Rapid Antigen Test 2.0 Nasal are considered useful and efficient RATs for the detection of SARS-CoV-2, particularly during the current Omicron subvariants wave.

## Introduction

Coronaviruses, members of the *Coronaviridae* family, are positive-strand RNA viruses with genomic RNA about 27–32 Kb in size. Most coronaviruses are enveloped and possess capped and polyadenylated genomic RNA. This group of viruses has been identified in mice, rats, chickens, turkeys, swine, dogs, cats, rabbits, horses, cattle, and humans. Coronaviruses cause diverse severe diseases, including gastroenteritis and respiratory tract-related diseases [1, 2]. Since first identified in the 1960s [3], multiple coronaviruses have been identified in humans. Among the identified human coronaviruses (hCoVs), the genus betacoronavirus (β-CoV), which contains four lineages (A, B, C, and D), is of the greatest clinical significance concerning

the study design, data collection, interpretation, or decision to submit the manuscript for publication.

**Competing interests:** The authors have declared that no competing interest exist.

humans. The β-CoVs that have raised concerns worldwide include HCoV-229E, HCoV-NL63, HCoV-OC43, HCoV-HKU-1, severe acute respiratory syndrome coronavirus (SARS-CoV), and middle east respiratory syndrome-related coronavirus (MERS-CoV) [4–6].

By the end of 2019, a new human coronavirus, known as severe acute respiratory syndrome coronavirus 2 (SARS-CoV-2), was first identified in Wuhan, China; it has rapidly spread worldwide since then, becoming a global pandemic [7]. Symptoms of the SARS-CoV-2 infection vary from mild, such as fever, cough, sore throat, and headache, to severe, such as pneumonia and respiratory distress syndrome, sometimes leading to death. Following the first outbreak of the COVID-19 pandemic, multiple variants of concern (VOCs) of SARS-CoV-2 were reported, including the Alpha (B.1.1.7), Beta (B.1.351), Gamma (P.1), and Delta (B.1.617.2) variants that were originally identified in the UK, South Africa, Brazil, and India, respectively [8, 9]. On November 9, 2021, another new variant of SARS-CoV-2 was identified in South Africa; this variant was later named Omicron [10]. Since then, Omicron was also identified in other countries, including the Netherlands, the USA, the UK, Canada, Australia, Japan, and France [11]. On 26 November 2021, the World Health Organization (WHO) announced Omicron as a VOC due to the presence of several mutations that may have an impact on disease transmissibility, severity, and ability to escape immune responses [12]. The Omicron variant was reported to cause a significantly lower proportion of moderate or severe diseases that lead to the death of the infected individuals, compared to the Delta variant [13]. Nevertheless, the Omicron variant has become a global urgent public health alert due to its high contagiousness and vaccine-evading properties [14]. Furthermore, the Omicron variant, remarkably, has evolved into multiple subvariants including BA.1, BA.2., BA.3, BA.4, and BA.5 [14, 15]. Following the initial spread of Omicron, BA.1 and BA.2 became the predominant subvariants, replacing the Delta variant [16]. However, since early 2022, BA.4 and BA.5 had become the predominant subvariants and have been classified by the WHO as VOC-subvariant under monitoring [17–20]. Owing to Omicron's enhanced transmissibility, the number of infected individuals exceeded those caused by the Delta variant or other VOCs [21]. Moreover, the emergence of evidence of the Omicron's ability, particularly that of the BA.4 and BA.5 subvariants, to evade vaccine-induced immunity placed the world on high alert regarding the Omicron variant.

To address this issue, numerous countermeasure strategies have been implemented, including the rapid identification and isolation of infected individuals. The rapid detection method, e.g., the rapid antigen test (RAT), to identify mutant viruses is crucial for these purposes. However, among the SARS-CoV-2 RATs that are currently available, many showed impaired performance for COVID-19 diagnosis when used for the detection of the Omicron variant [22]. In October 2022, Roche Diagnostics launched SARS-CoV-2 rapid antigen tests with an enhanced ability of antibodies in capturing the antigen, hence believed to be more effective in the detection of SARS-CoV-2 regardless of the variants. The SARS-CoV-2 Rapid Antigen Test 2.0 and SARS-CoV-2 Rapid Antigen Test 2.0 Nasal (Roche Diagnostic, Mannheim, Germany) manufactured by SD Biosensor (Suwon, South Korea), are novel RATs used to qualify SARS-CoV-2 antigen (nucleocapsid protein) by immune chromatography from a nasopharyngeal or nasal swab of patients suspected with SARS-CoV-2 infection. These RATs were equipped with the anti-SARS-CoV-2 antibody, anti-chicken IgY, anti-SARS-CoV-2 antibody-gold conjugate, and the purified chicken IgY-gold conjugate [23, 24]. Given that the SARS-CoV-2 Rapid Antigen Test 2.0 and SARS-CoV-2 Rapid Antigen Test 2.0 Nasal were launched during the Omicron wave, a thorough evaluation of these RATs particularly for the detection of the Omicron variant and its subvariants is crucial.

In this study, we evaluated the clinical sensitivity of the SARS-CoV-2 Rapid Antigen Test 2.0 and SARS-CoV-2 Rapid Antigen Test 2.0 Nasal to detect the SARS-CoV-2, particularly the

Omicron variant and its subvariants from the nasopharyngeal and nasal swab specimens. Additionally, we identified the Omicron subvariants from RAT-positive specimens by Omicron real-time PCR (Omicron RT-PCR) and whole-genome sequencing (WGS). The sensitivity of the RATs was determined by comparing the results from RAT with the ones from the RT-PCR, Omicron RT-PCR, and WGS.

## Materials and methods

### Study design

In total, 56 individuals who visited the Gyeongsang National University Changwon Hospital (GNUCH) from November to December 2022, with either mild or severe symptoms within seven days from the symptom onset (DSO), participated in this study. The participant's recruitment, sample collection, and assessments were conducted between November to December 2022. Individuals under 19 years of age or who did not provide informed consent were excluded from this study. All participating individuals agreed to this study and submitted their written informed consent. Only the researchers that conducted the experiments have access to information that could identify individual participants during or after data collection. The study protocol was approved by the institutional review board of Gyeongsang National University Changwon Hospital (IRB No. 2022–10–009). Two nasopharyngeal and one nasal swab specimens were collected from each of the participants. The nasopharyngeal swab specimens that were designated for reverse-transcription real-time PCR (RT-PCR) and whole-genome analyses were stored in the viral transport medium (VTM) (Noble Bio, Hwaseong, Korea) at -80˚C before being used (Fig 1).

### Clinical performance of rapid antigen tests (RATs)

The RATs were conducted with the SARS-CoV-2 Rapid Antigen Test 2.0 and SARS-CoV-2 Rapid Antigen Test 2.0 Nasal following the manufacturer's instructions. Briefly, after sample collection, a swab in the buffer tube was moved around in a circle in place, squeezed into the tube wall, and four drops of the reaction mixture were applied to the device. Subsequently, the antigen for COVID-19 in the sample mixture binds to the antibodies embedded in the control (C) and test (T) lines and leads to the development of a band of color(s). The band(s) of color that appeared after 15 min on the C and T were interpreted as a test result (Fig 2).

### Reverse transcription real-time polymerase chain reaction (RT-PCR)

The STANDARD M nCoV Real-time Detection kit (SD Biosensor, Cat. No 11NCO10) and the PowerCheck™ SARS-CoV-2 Omicron Variants Real-time PCR kit (KogenBiotech, Seoul, Korea, Cat. No R6922Q) were used for molecular detection of SARS-CoV-2 from the specimens. The STANDARD M nCoV Real-time Detection kit, hereinafter referred to as RT-PCR, enables the detection of SARS-CoV-2 by identifying the *RdRp* and *E* genes from the specimens. Meanwhile, the PowerCheck™ SARS-CoV-2 Omicron Variants Real-time PCR kit, hereinafter referred to as Omicron RT-PCR, enables the detection of the mutated *RBD* in the *S* gene, which enables simultaneous and individual discrimination for the subvariants of Omicron. The assessments were conducted according to the manufacturer's instructions. The results were interpreted as positive if the cycle threshold (Ct) value was within the cutoff values and negative if they were outside the cutoff or if there was no amplification.

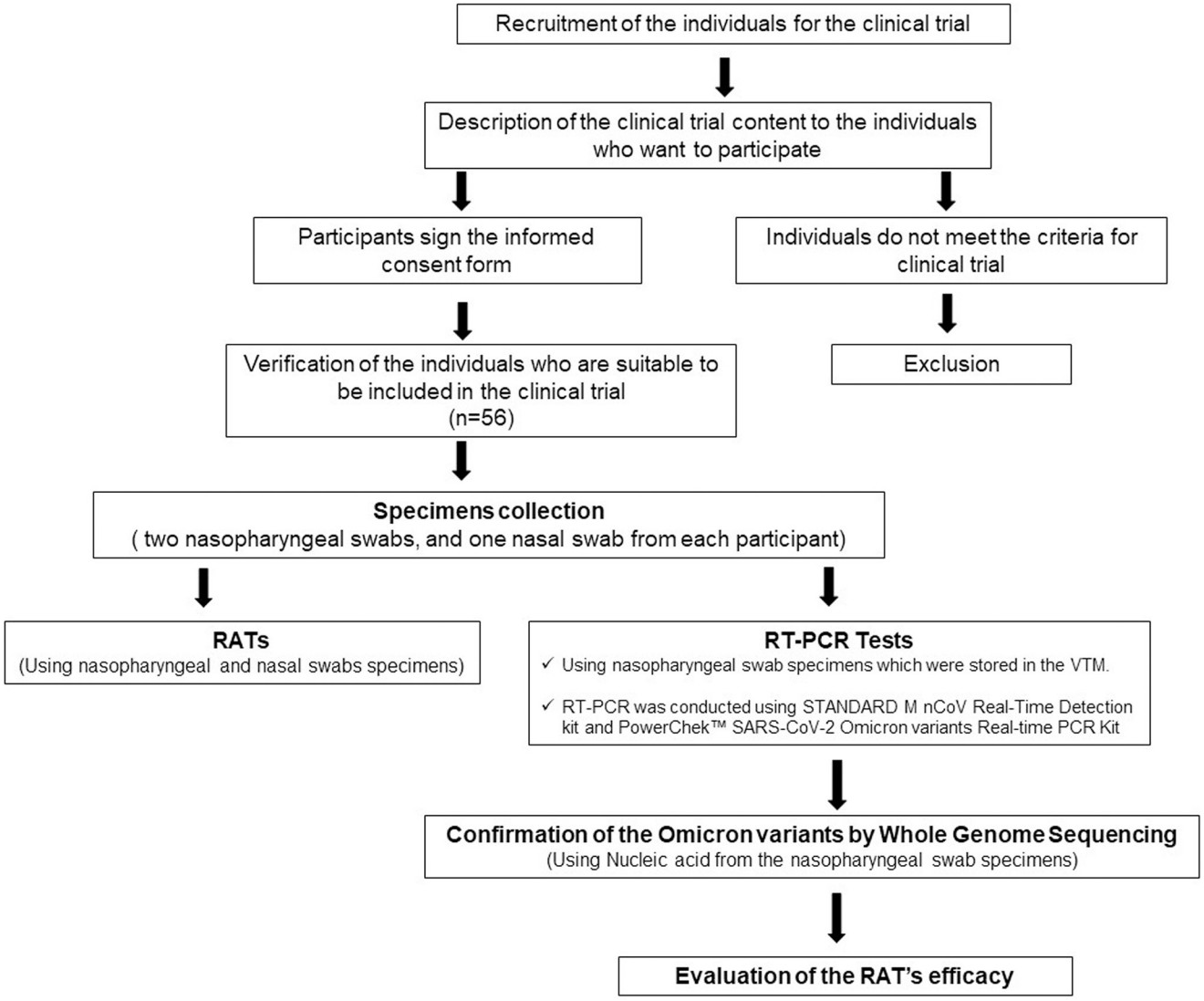

**Fig 1. Diagram representing the study design.** Specimens were collected from the nasopharyngeal and nasal swabs of individuals that met the requirements to participate in the study. The clinical performance evaluation of the SARS-CoV-2 Rapid Antigen Test 2.0 and SARS-CoV-2 Rapid Antigen Test 2.0 Nasal was conducted following the sample collection. The RT-PCRs were conducted for molecular detection of SARS-CoV-2 and identification of the Omicron subvariants. Furthermore, the remaining total RNA that was extracted from the nasopharyngeal swab samples was sent to an external professional institution for whole-genome sequencing (WGS) analysis. Finally, the RAT, RT-PCRs, and WGS results were compared to determine the clinical efficacy of the RATs.

## Whole-genome sequencing (WGS)

We extracted the total RNA from the nasopharyngeal specimens and sent it to an external professional institution (Theragen Bio, Seongnam, Korea) for WGS analysis. Briefly, the cDNA was synthesized from the total viral RNA by QIAseq SARS-CoV-2 Primer Panel (QIAGEN, Hilden, Germany, Cat. No 333895). Following cDNA synthesis, primer pools based on high-fidelity multiplex PCR were used to prepare two pools of 400 bp QIAseq SARS-CoV-2 Primer Panel amplicons. Two enriched pools per sample were put into a single tube and were purified. Subsequently, the DNA library was constructed. Target-enriched samples from the QIAseq SARS-CoV-2 Primer Panel were fragmented by enzymatic shearing. Fragments of 250 bp were ligated to Illumina's adapters and PCR-amplified. These adapters contain sequences essential

**Fig 2. The SARS-CoV-2 Rapid Antigen Test 2.0 and SARS-CoV-2 Rapid Antigen Test 2.0 Nasal results.** The color line also appears in the control line C and the test line T within 15 minutes when the sample mixture contains an antigen of SARS-CoV-2, indicating that the test is positive for SARS-CoV-2. One line on the C marker indicates that the test is negative.

for binding dual-barcoded libraries to a flow cell for sequencing. An appropriate size for the final library is 300–500 bp. The resulting purified libraries were applied to an Illumina flow cell for cluster generation and were sequenced using 150 bp paired-end reads on an Illumina NovaSeq 6000 (Illumina, San Diego, CA, Cat. No 20012850) sequencer by following the manufacturer's protocol. Subsequently, the obtained sequences were compared with the sequences database in the global initiative for sharing all influenza data (GISAID, https://gisaid.org/), and the mutation database was constructed at the "mutation and lineage-specific mutation database" site for classifying the mutation that occurs. The Omicron subvariants were determined through "variant calling" in the WGS analysis, which is the process by which the variants were identified according to the sequence data deposited in the database (GISAID). Briefly, the quality of the sequencing data in the format of Fastq was checked using the FastQC (v0.10.1) program, meanwhile, the Base Quality, Duplication level, and GC content were visually checked. Subsequently, the trimming procedure was employed to filter the reads to remove the low-quality sequences. The sickle (v.1.3.3) program was used for the entire filtering process. Subsequently, the reads were aligned to the SARS-CoV-2 reference using BWA (B0.7.17) program, and the duplication reads generated through PCR were removed using GATK (v.4.0.2.1) program. Finally, the stand_call_conf—30.0, stand_emit_conf -10.0, dcov—1,000 options were applied to find mutations within the sequence which already clear out from any duplication, and the variant analysis was performed using GATK (v.4.0.2.1). Further, the annotation of mutation from GATK (v.4.0.2.1) was finalized using the SnpEff (V.4.1) program and the SARS-CoV-2 strain of each sample was determined after comparing it with the

sequences database in the global initiative for sharing all influenza data (GISAID, https://gisaid.org/) and with the mutation databases that were obtained from the lineage-specific mutation information database provided by the outbreak.info [25], CoVariants [26], and COVID CG [27, 28]. The result is referred to as "uncertain" when the lineage-specific variants are not detected, or only mutations shared by two or more lineages are called from the "variant calling".

## Phylogenetic analysis

To find a relationship between the subvariants of Omicron that were identified from the specimens, we aligned and analyzed the spike protein sequences of the identified subvariants by using ClustalW and generated a phylogenetic tree using the neighbor-joining and bootstrap methods implemented in the molecular evolutionary genetic analysis version X (MEGA X, Pennsylvania State University, PA) [29]. In this study, the spike protein was chosen for the phylogenetic analysis, considering that the mutation of SARS-CoV-2 variants (including their subvariants) occurred more likely in the spike protein.

## Statistical analysis

All tests were performed in a blinded manner by two researchers. The diagnostic sensitivity of the SARS-CoV-2 Rapid Antigen Test 2.0 and SARS-CoV-2 Rapid Antigen Test 2.0 Nasal was determined using a comparison analysis against RT-PCR, Omicron RT-PCR, and WGS results. We performed all statistical analyses using SAS software version 9.4 (SAS Institute Inc., Cary, NC).

## Results

### Clinical sensitivity of the SARS-CoV-2 RATs according to RT-PCR

We assessed the clinical sensitivity of the SARS-CoV-2 Rapid Antigen Test 2.0 and SARS-CoV-2 Rapid Antigen Test 2.0 Nasal by comparing the results obtained with those of the RT-PCR gold standard for the detection of SARS-CoV-2. Both SARS-CoV-2 Rapid Antigen Test 2.0 and SARS-CoV-2 Rapid Antigen Test 2.0 Nasal demonstrated good performance when being used to assess the nasopharyngeal and nasal swab samples from the suspected individuals, with a sensitivity of 92.86% (95% CI 82.71%– 98.02%) (Table 1).

**Table 1. Sensitivity of the SARS-CoV-2 Rapid Antigen Test 2.0 and the SARS-CoV-2 Rapid Antigen Test 2.0 Nasal compared to the STANDARD M nCoV Real-Time Detection kit in detecting the SARS-CoV-2 from the nasopharyngeal and nasal specimens, respectively.**

| | | STANDARD M nCoV Real-Time Detection kit | | Total |
| --- | --- | --- | --- | --- |
| | | Positive | Negative | |
| SARS-CoV-2 Rapid Antigen Test 2.0 | Positive | 52 | 0 | 52 |
| | Negative | 4 | 0 | 4 |
| SARS-CoV-2 Rapid Antigen Test 2.0 Nasal | Positive | 52 | 0 | 52 |
| | Negative | 4 | 0 | 4 |
| | Total | 56 | 0 | 56 |

Sensitivity of both RATs: 92.86%, (95% CI 82.71%– 98.02%)

Abbreviation: CI, confidence interval

## Clinical sensitivity of the SARS-CoV-2 RATs according to the Ct value of the *RdRp* and *E* genes

Further, we analyzed the sensitivity of the SARS-CoV-2 RATs by restricting the Ct value for *RdRp* and *E* genes of the initial RT-PCR positive-diagnosed specimens into three groups, Ct<15, 15≤Ct<25, and 25≤Ct≤35. The sensitivity of both RATs for specimens with a Ct value for the *RdRp* gene of <15 was 82.18% (95% CI 48.55%– 97.86%). This value was increased to 97.05% (95% CI 83.46%– 99.84%) for a 15≤Ct<25 Ct. However, when the range of Ct value was increased to 25≤Ct≤35, the sensitivity for both RATs was decreased to 91.09% (95% CI 57.11%– 99.5%) (Table 2). Correspondingly, the sensitivity of RATs compared to the Ct value for the *E* gene produced an equivalent result to the one compared to the Ct value for *RdRp* (Table 2).

## Clinical sensitivity of the SARS-CoV-2 RATs compared to the Omicron RT-PCR

The sensitivity of the SARS-CoV-2 RATs compared to the Omicron RT-PCR was up to 94.23% (95% CI 83.07%– 98.49%) for both SARS-CoV-2 Rapid Antigen Test 2.0 and SARS-CoV-2 Rapid Antigen Test 2.0 Nasal (Table 3). Furthermore, among 56 specimens that were analyzed by Omicron RT-PCR, 43 specimens (76.8%) were identified as Omicron subvariant BA.4 or BA.5, and nine (16.1%) were identified as subvariant BA.2.75. BA.4 and BA.5 are indistinguishable by Omicron RT-PCR. Meanwhile, the negative results were present either due to the absence of the Omicron in the specimens (three specimens), or due to the Omicron RT-PCR test that provided neither positive nor negative results, although the test was repeated (one specimen) (Table 3).

## Clinical sensitivity of the SARS-CoV-2 RATs compared to whole-genome sequencing (WGS)

We also compared the result from the RATs assay with that from WGS. In this study, we assessed 55 specimens for WGS, comprised of 52 confirmed positive by both RATs and Omicron PCR, and three confirmed positive by only Omicron PCR. One specimen was excluded from WGS analysis due to the negative result from both RATs and Omicron PCR tests. The sensitivity analysis of the SARS-CoV-2 RATs compared to the WGS results was up to 97.95% (95% CI 87.76%– 99.89%) for both the SARS-CoV-2 Rapid Antigen Test 2.0 and SARS-CoV-2 Rapid Antigen Test 2.0 Nasal (Table 4).

**Table 2. Sensitivity of the SARS-CoV-2 Rapid Antigen Test 2.0 and SARS-CoV-2 Rapid Antigen Test 2.0 Nasal according to Ct value for *RdRp* gene and *E* gene.**

| SARS-CoV-2 Rapid Antigen Test 2.0 | | | | SARS-CoV-2 Rapid Antigen Test 2.0 Nasal | | | |
|---|---|---|---|---|---|---|---|
| Ct Value | N | Sensitivity | 95% CI | N | Sensitivity | 95% CI | |
| *RdRp* gene | | | | | | | |
| Ct<15 | 11 | 82.18% | 48.55%– 97.86% | 11 | 82.18% | 48.55%– 97.86% | |
| 15≤Ct<25 | 34 | 97.05% | 83.46%– 99.84% | 34 | 97.05% | 83.46%– 99.84% | |
| 25≤Ct≤35 | 11 | 91.09% | 57.11%– 99.50% | 11 | 91.09% | 57.11%– 99.50% | |
| *E* gene | | | | | | | |
| Ct<15 | 11 | 85.15% | 54.62%– 97.28% | 11 | 85.15% | 54.62%– 97.28% | |
| 15≤Ct<25 | 36 | 97.22% | 84.96%– 99.85% | 36 | 97.22% | 84.96%– 99.85% | |
| 25≤Ct≤35 | 9 | 89.88% | 51.70%– 99.41% | 9 | 89.88% | 51.70%– 99.41% | |

Abbreviations: Ct, the cycle of threshold; N, the number of positives specimens based on rapid Ag tests; CI, confidence interval

**Table 3. Performance of the SARS-CoV-2 Rapid Antigen Test 2.0 and SARS-CoV-2 Rapid Antigen Test 2.0 Nasal compared to the PowerCheck™ SARS-CoV-2 Omicron Variants Real-time PCR kit (N = 56).**

| | | PowerChek™ SARS-CoV-2 Omicron variants Real-time PCR | | | | PowerChek™ SARS-CoV-2 Omicron variants Real-time PCR | | |
|---|---|---|---|---|---|---|---|---|
| | | Positive | Not detected* | N/A** | | | | |
| SARS-CoV-2 Rapid Antigen Test 2.0 and Test 2.0 Nasal | Positive | 49 | 2 | 1 | Subvariants | BA.4 or BA.5 | 43 | |
| | | | | | | BA.2.75 | 9 | |
| | Negative | 3 | 1 | 0 | | Not detected* | 3 | |
| | | | | | | N/A** | 1 | |
| Sensitivity: 94.23% (95% CI 83.07%– 98.49%) | | | | | | | | |

* Omicron variant was not detected.

** N/A (not available): Repeated tests yielded neither positive nor negative results.

Abbreviation: CI, confidence interval

Subvariants analysis by WGS (S1 Fig) demonstrated that from total of 55 specimens, 36 specimens (65.5%) were positive for Omicron subvariant BA.5, one specimen for either Omicron subvariant BA.5 or BA.4, one specimen for either subvariant BA.4 or BA.2.75, 11 specimens (20%) for subvariant BA.2.75, and six specimens (10.9%) for an uncertain variant of SARS-CoV-2 (Table 4). The six specimens identified as "uncertain" showed a low value of the variables that were being assessed in the WGS analysis, hence making them difficult to be distinguished into specific lineages (S1 Fig).

Furthermore, a phylogenetic analysis of the spike protein of several subvariants of Omicron revealed that BA.4, BA.5, and BA.2.75 share a lot of similarities in their S protein sequences; thus, they are located in close branches of the phylogenetic tree (Fig 3).

## Discussion

In comparison to the prior SARS-CoV-2 variants, the Omicron showed a significant number of mutations [30], with a greater rate of transmissibility [31, 32] and the ability to evade the efficacy of the currently administered COVID-19 vaccines or monoclonal antibody therapies [33, 34]. As part of the SARS-CoV-2 control measures, particularly regarding Omicron and its subvariants, rapid detection by RATs plays a critical role in the prevention of the further

**Table 4. Performance of the SARS-CoV-2 Rapid Antigen Test 2.0 and SARS-CoV-2 Rapid Antigen Test 2.0 Nasal compared to whole-genome sequencing (N = 55).**

| | | Whole Genome Sequencing | | | Whole Genome Sequencing | | |
|---|---|---|---|---|---|---|---|
| | | Positive | Uncertain* | | | | |
| SARS-CoV-2 Rapid Antigen Test 2.0 and Test 2.0 Nasal | Positive | 48 | 4 | Subvariants | BA.5 | 36 | |
| | | | | | BA.4 or BA.5** | 1 | |
| | | | | | BA.2.75 or BA.4** | 1 | |
| | | | | | BA.2.75 | 11 | |
| | Negative | 1 | 2 | | Uncertain* | 6 | |
| | Excluded** | 1 | | | Excluded*** | 1 | |
| Sensitivity: 97.95% (95% CI 87.76%– 99.89%) | | | | | | | |

*Uncertain refers to the samples identified as SARS-CoV-2 Omicron variant but difficult to determine into accurate subvariant.

**It was difficult to distinguish between these two subvariants.

***The sample was excluded from whole genome sequencing analysis because the Omicron real-time PCR nor RATs gave positive results.

Abbreviation: CI, confidence interval

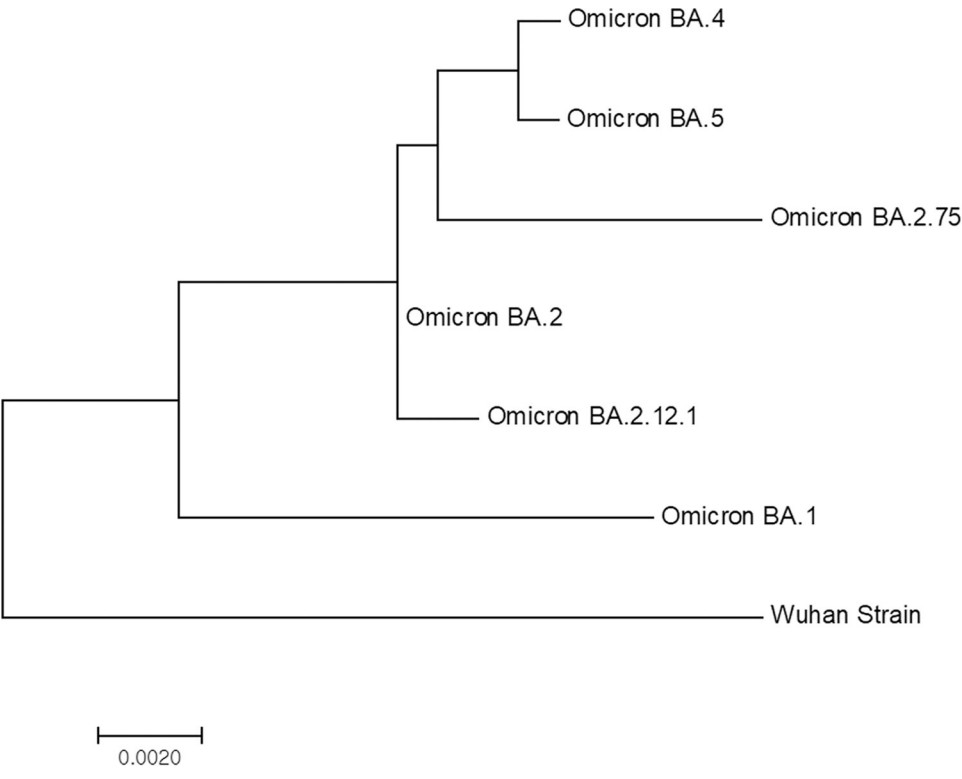

**Fig 3. Phylogenetic analysis of the spike protein of the Omicron subvariants.** Omicron subvariants BA.4 and BA.5 are located in the same branch, confirming the high similarity between these two subvariants, whereas the subvariant BA.2.75 shared the same node as subvariants BA.4 and BA.5, suggesting that these subvariants evolved from a common ancestor.

spread of the Omicron variant and helps the physician to make a proper decision regarding treatment for infected individuals.

However, the current commercially available SARS-CoV-2 RATs were reported to have a diverse performance for the detection of the Omicron variant. Evaluation of the total of 120 samples using six commercial RATs demonstrated a heterogenicity of sensitivity among the RATs. The sensitivity of the RATs for detecting the Omicron variant across a wide range of cycles of threshold (Ct) values, with a median of 21.3, ranged from 69.6% to 78.3%. Subsequently, when being used to assess samples with lower viral load (Ct > 25, corresponding to <4.9 $\log_{10}$ copies/mL), the sensitivity of the RATs was decreased significantly, to as low as 0.0% to 23.1% [35]. Correspondingly, analysis of five SARS-CoV-2 RATs by another group demonstrated true positive rates in detecting Omicron variants between 44.7% to 91.1% for high viral load (Ct <25); these dropped to 8.7% to 22.7% in the analysis of samples with intermediate Ct values (Ct of 25–30) [36].

In this study, we assessed the clinical sensitivity of two novel SARS-CoV-2 RATs for the detection of SARS-CoV-2, particularly Omicron and its subvariants, from the nasopharyngeal (SARS-CoV-2 Rapid Antigen Test 2.0) and nasal swabs specimens (SARS-CoV-2 Rapid Antigen Test 2.0 Nasal). Sample collection and analysis were conducted between November and December 2022; the Omicron variant was reported as the predominant variant in January 2022 in South Korea [37, 38]. In our study, both the SARS-CoV-2 Rapid Antigen Test 2.0 and the SARS-CoV-2 Rapid Antigen Test 2.0 Nasal demonstrated high sensitivity in detecting the Omicron variant in the samples from the suspected individuals, regardless of the specimen

collection sites or ranges of the Ct values. However, when the analysis was conducted by restricting the Ct value, we found that two of 11 samples with a Ct value of <15 produced a negative result, making the sensitivity of this group appear lower (82.18%) than the group of specimens with a Ct value of 15≤Ct<25 (97.05%). We assumed this condition was related to technical failure during the assessment by RATs. Our inability to repeat the RAT assessment for confirmation of these two samples due to the insufficient amounts of the remaining samples is considered a limitation. Nevertheless, the overall sensitivity of both RATs is higher than the sensitivity required by the WHO [39]. Moreover, our result is in agreement with a recent study that reported high performance of SARS-CoV-2 Rapid Antigen Test 2.0 and SARS-CoV-2 Rapid Antigen Test 2.0 Nasal [40], suggesting that these RATs are considered promising detection tools of major variants of concern of SARS-CoV-2.

Further, we analyzed the Omicron subvariants by Omicron RT-PCR analysis and found that BA.4/BA.5 and BA.2.75 were the commonly identified subvariants. The Omicron RT-PCR, however, could not distinguish between BA.4 and BA.5; therefore, the result appears as BA.4/BA.5. In our analysis, BA.4/BA.5 appeared as the predominant subvariant (78.6%). Subsequently, we also evaluated the clinical sensitivity of the RATs compared to the Omicron RT-PCR result, finding a high sensitivity, comparable to the one obtained from the initial RT-PCR.

Furthermore, we also confirmed the identified subvariants by WGS. The sequencing analysis demonstrated that the Omicron BA.5 was the dominant subvariant (65.5%). This result is in concordance with the actual condition in South Korea, where the subvariant BA.5 was dominant during the study period (November–December 2022) [37, 41, 42]. Furthermore, WGS confirmed that 20% of specimens were positive for subvariant BA.2.75. The subvariant BA.2.75 was first confirmed in July 2022 in South Korea [43]—two months later than the BA.5 which was first discovered in May and became dominant when the study was conducted (November–December 2022) [44]. Given its high transmissibility and ability to evade immunity, the subvariant BA.2.75 is also likely to cause breakthrough infections or reinfections as BA.5; hence, the number of individuals infected by BA.2.75 in this study appeared high, constituting the second most dominant subvariant after BA.5. Furthermore, among WGS results, two specimens showed the presence of subvariants that were difficult to identify as either BA.4/BA.5 or BA.2.75/BA.4. The number of specimens with indistinguishable subvariants was much smaller than the number assessed by Omicron RT-PCR, suggesting that WGS provides more reliable and accurate identification of the Omicron subvariants than Omicron RT-PCR.

The BA.4 and BA.5 subvariants have a high similarity of spike protein mutation with BA.2 [17, 20]. In addition, given that BA.2.75 evolved from BA.2, we assumed that the difficulties distinguishing between these variants are due to their sequence similarities. Therefore, we analyzed the spike protein sequences among Omicron subvariants and generated a phylogenetic tree. As expected, the phylogenetic analysis of the spike proteins from subvariants BA.4, BA.5, and BA.2.75 demonstrated a close relationship that corresponds with the high similarities of their spike protein sequence.

The performance of RATs for the detection of SARS-CoV-2, particularly the Omicron and its subvariants is influenced by several factors, including the viral genetics and the design of the test devices. Hence, each RAT is affected differently by viral mutation due to the difference in the inherent design of each RAT. Apart from the studies that reported the decline of the RAT for Omicron detection, other studies claimed that some of the currently available RATs are not affected by the emergence of Omicron [45–48], particularly when being used to assess the samples with high viral load. The DIAGNOVIR SARS-CoV-2 ultra-rapid antigen test, for example, was reported to demonstrate high performance in detecting most SARS-CoV-2 variants including Wuhan, Alpha (B1.17), Beta (B.1.351), Delta (B.1.617.2), and Omicron

(B.1.1529) [49]. A study on several currently available RAT kits demonstrated a similar level of sensitivity to Omicron and non-Omicron variants (On/Go, 76.4% and 71.0%; iHealth, 73.0%, and 71.0%; QuickVue, 84.3%, and 74.3%; BinaxNOW, 69.7%, and 71.0%; and InBios, 66.3% and 64.5%, respectively) [45]. However, most of these studies only evaluated the sensitivity and specificity for detecting Omicron B.1.1529 but not for detecting currently circulating subvariants [45, 46, 49]. A study reported that most of the Omicron subvariants accumulated amino acid substitutions in the N protein [50]. Given that the N protein is an important target protein for most of the currently available RATs, any mutation in this protein may affect the sensitivity of RATs [51]. Hence, the SARS-CoV-2 Rapid Antigen Test 2.0 and SARS-CoV-2 Rapid Antigen Test 2.0 Nasal, which enable rapid and accurate detection of SARS-CoV-2, particularly the currently circulating Omicron subvariants such as BA.4 and BA.5, are considered promising to assist in the control of the person-to-person or closed-community transmission of these subvariants.

Both SARS-CoV-2 Rapid Antigen Test 2.0 and SARS-CoV-2 Rapid Antigen Test 2.0 Nasal are intended for use in the laboratory or point-of-care setting by healthcare professionals, not for home/self-testing. This may be a limitation of these SARS-CoV-2 RATs compared to the currently available home/self-testing RATs. Additionally, the RATs used in this study were conducted on individuals with mild or severe symptoms within 7 DSO with median Ct values of 18.64 for *RdRp*. Hence, further study on the performance of SARS-CoV-2 Rapid Antigen Test 2.0 and SARS-CoV-2 Rapid Antigen Test 2.0 Nasal for asymptomatic individuals or symptomatic individuals with more than seven DSO and a wider range of Ct values may be necessary. Additionally, in this study, we only determined the sensitivity for Omicron-positive cases and did not analyze specificity for negative cases. Lastly, considering the identified subvariants in this study were limited to BA.4, BA.5, and BA2.75, which are major subvariants found in South Korea during the study period, clinical performance evaluation of these SARS-CoV-2 RATs for detection of other variants of SARS-CoV-2, may also be necessary.

Nevertheless, our results demonstrated an exceptional performance of SARS-CoV-2 Rapid Antigen Test 2.0 and SARS-CoV-2 Rapid Antigen Test 2.0 Nasal in detecting Omicron in the suspected individuals, regardless of its subvariants, making them useful and efficient for the detection of SARS-CoV-2, particularly during the current Omicron wave.

## Conclusions

The SARS-CoV-2 Rapid Antigen Test 2.0 and SARS-CoV-2 Rapid Antigen Test 2.0 Nasal demonstrated high sensitivity in detecting Omicron subvariants. Therefore, these RATs are considered promising to provide an alternative method that is effective for surveillance and control measure strategies to distinguish infected individuals from those who are not infected amid the Omicron wave.

## Supporting information

**S1 Fig. Whole genome sequencing analysis of the respiratory RNA samples from the individuals confirmed positive SARS-CoV-2 by SARS-CoV-2 RATs.** In, (A) the coverage depth of filtered raw data across reference sequences, (B) annotated variants that occur in the obtained sequences, (C) annotated categories of functionally important regions, and (D) mutation classes of sequence variants.
(TIF)

## Acknowledgments

The authors wish to thank In-Young Kang, Sung-Hwa Park, and Soomin Kim for their assistance during the experiment.

## Author Contributions

**Conceptualization:** Sunjoo Kim.

**Data curation:** Kristin Widyasari.

**Formal analysis:** Kristin Widyasari.

**Funding acquisition:** Sunjoo Kim.

**Methodology:** Kristin Widyasari.

**Supervision:** Sunjoo Kim.

**Writing – original draft:** Kristin Widyasari.

**Writing – review & editing:** Sunjoo Kim.

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
