## [Decision Letter · Decision Letter 0]

7 Jun 2023

PONE-D-23-10862Efficacy of novel SARS-CoV-2 rapid antigen tests in the era of omicron outbreakPLOS ONE

Dear Dr. Kim,

Thank you for submitting your manuscript to PLOS ONE. After careful consideration, we feel that it has merit but does not fully meet PLOS ONE’s publication criteria as it currently stands. Therefore, we invite you to submit a revised version of the manuscript that addresses the points raised during the review process.

We look forward to receiving your revised manuscript.

Kind regards,

Ziwei Ye, Ph.D

Academic Editor

PLOS ONE

Journal Requirements:

Reviewers' comments:

Reviewer's Responses to Questions

**Comments to the Author**

1. Is the manuscript technically sound, and do the data support the conclusions?

Reviewer #1: Yes

Reviewer #2: Yes

2. Has the statistical analysis been performed appropriately and rigorously? 

Reviewer #1: Yes

Reviewer #2: Yes

3. Have the authors made all data underlying the findings in their manuscript fully available?

Reviewer #1: No

Reviewer #2: Yes

4. Is the manuscript presented in an intelligible fashion and written in standard English?

Reviewer #1: Yes

Reviewer #2: Yes

5. Review Comments to the Author

Reviewer #1: The current manuscript is strict-forward in determining the sensitivity of SARS-CoV-2 Rapid Antigen Test 2.0 and SARS-CoV-2 Rapid Antigen Test 2.0 Nasal to Omicron variant. Not many major concerns exist. Some clarification and elaboration can improve the manuscript before publication. Thank you.

1. It will be better to mention earlier that SARS-CoV-2 Rapid Antigen Test 2.0 and SARS-CoV-2 Rapid Antigen Test 2.0 Nasal are designed and intended to detect Omicron variant. This background information is unclear until line 294. What design was made for these two kits? E.g. specifying what specific detection antibodies are included in the kit will help improve the delivered knowledge of the paper.

2. Please clarify the definition of “uncertain” in table 4. Does it mean it is not Omicron, uncertain to be SARS-CoV-2 or negative?

3. Analysis method of WGS was not mentioned clearly (line 147-161). What is the criteria for WGS to define Omicron subtypes and uncertain.

4. What is the significance of having a RAT kit to detect specifically Omicron variant? Strength and drawback can be discussed. How about a pan-SARS-CoV-2 strains RAT kit e.g. DIAGNOVIR (PMID: 36932107). Possible and necessary to include few more detection lines on a RAT kit to distinguish Omicron variants?

5. Some studies claim currently available RAT kits are not affected by the emergence of Omicron variant (PMID: 36519888, PMID: 36215709, PMID: 36798414, PMID: 35217106). Discussion can be provided to compare the claim of the current manuscript to gain a boarder view for whether it is a real case that currently available RAT kits are not sensitive enough to Omicron variant.

6. WGS sequencing result was not available from the manuscript although authors declare Yes for all data are fully available without restriction.

7. Line 48, HKUI can be converted to HKU-1.

8. Line 80-81, reference citation is required.

Reviewer #2: This is a timely manuscript.

1. it is not audience friendly to place the figure legends under Materials and Methods.

2. More details about the RT-PCR and WGS shall be provided, including commercial sources and Cat#.

6. PLOS authors have the option to publish the peer review history of their article (what does this mean?). If published, this will include your full peer review and any attached files.

Reviewer #1: **Yes: **Cheung Pak Hin Hinson

Reviewer #2: No

---

## [Author Response · Author response to Decision Letter 0]

18 Jun 2023

The authors would like to thank the Reviewers for their specific and helpful comments on the manuscript. The authors have carefully taken the comments into consideration and have made revisions to the manuscript to address reviewers’ and editors’ concerns.

Responses to the Reviewer are reported as follows:

Response: 

We have ensured that the file naming has met PLOS ONE's style requirements

2. We note that you have included the phrase “data not shown” in your manuscript. Unfortunately, this does not meet our data-sharing requirements. PLOS does not permit references to inaccessible data. We require that authors provide all relevant data within the paper, Supporting Information files, or in an acceptable, public repository. Please add a citation to support this phrase or upload the data that corresponds with these findings to a stable repository (such as Figshare or Dryad) and provide URLs, DOIs, or accession numbers that may be used to access these data. Or, if the data are not a core part of the research being presented in your study, we ask that you remove the phrase that refers to these data.

Response:

We have added the mentioned data to Table 2.

Pages 11-12, Lines: 219-225, Table 2 (E gene)

Reviewers Comments to the Author

Reviewer #1: 

The current manuscript is strict-forward in determining the sensitivity of SARS-CoV-2 Rapid Antigen Test 2.0 and SARS-CoV-2 Rapid Antigen Test 2.0 Nasal to Omicron variant. Not many major concerns exist. Some clarification and elaboration can improve the manuscript before publication. Thank you.

1. It will be better to mention earlier that SARS-CoV-2 Rapid Antigen Test 2.0 and SARS-CoV-2 Rapid Antigen Test 2.0 Nasal are designed and intended to detect the Omicron variant. This background information is unclear until line 294. What design was made for these two kits? E.g. specifying what specific detection antibodies are included in the kit will help improve the delivered knowledge of the paper.

Response:

 We have confirmed to the manufacturer that SARS-CoV-2 Rapid Antigen Test 2.0 and SARS-CoV-2 Rapid Antigen Test 2.0 Nasal are not specifically designed for only detecting the Omicron variant but for all SARS-CoV-2 variants. However, given that there are some modifications, the manufacturer claimed that the antibodies have a better ability to capture the SARS-CoV-2 antigen regardless of the variant. 

We have added this information on pages 4-5 lines: 81-92.

“In October 2022, Roche Diagnostics launched SARS-CoV-2 rapid antigen tests with an enhanced ability of antibodies in capturing the antigen, hence believed to be more effective in the detection of SARS-CoV-2 regardless of the variants.”

We also have modified the sentences in line 294 to avoid any misunderstanding regarding the RATs.

Page 16, lines 309-312. 

“In this study, we assessed the clinical sensitivity of novel SARS-CoV-2 RATs for the detection of SARS-CoV-2, particularly Omicron and its subvariants, from the nasopharyngeal (SARS-CoV-2 Rapid Antigen Test 2.0) and nasal swabs specimens (SARS-CoV-2 Rapid Antigen Test 2.0 Nasal).”

2. Please clarify the definition of “uncertain” in Table 4. Does it mean it is not Omicron, uncertain to be SARS-CoV-2 or negative?

Response:

 We have added the definition of “uncertain” in Table 4 legends. 

“Uncertain refers to the samples identified as SARS-CoV-2 Omicron variant but difficult to determine into accurate subvariant.”

3. Analysis method of WGS was not mentioned clearly (lines 147-161). What are the criteria for WGS to define Omicron subtypes and uncertain?

Response:

 The Omicron subvariant was determined according to the mutations that occur in the sequence after comparing the sequence with the database in GISAID. In short, the obtained sequences were compared with the sequence data that were stored in the GISAID, and the mutation database was constructed for classifying the mutation that occurs. The Omicron subvariants were determined through variant calling in the WGS analysis. The result is referred to as “uncertain” when the lineage-specific variants are not detected, and only mutations shared by two or more lineages are called from the “variant calling”.

Page 8, Lines: 169-174. 

4. What is the significance of having a RAT kit to detect specifically Omicron variant? Strengths and drawbacks can be discussed. How about a pan-SARS-CoV-2 strains RAT kit e.g. DIAGNOVIR (PMID: 36932107)? Possible and necessary to include a few more detection lines on a RAT kit to distinguish Omicron variants?

Response:

 We have clarified that the SARS-CoV-2 Rapid Antigen Test 2.0 and SARS-CoV-2 Rapid Antigen Test 2.0 Nasal are not specifically only detecting the Omicron from the samples, but all SARS-CoV-2 regardless of the variants including the Omicron and its subvariants due to the modification on the binding ability of the antibody used in the RAT’s design. (pages 4-5 lines: 81-92). 

 However, as suggested by the reviewer, we added a discussion of the strengths of SARS-CoV-2 Rapid Antigen Test 2.0 and SARS-CoV-2 Rapid Antigen Test 2.0 Nasal in detecting Omicron compared to other RAT kits. 

Pages 18-19, Lines: 358-379

5. Some studies claim currently available RAT kits are not affected by the emergence of the Omicron variant (PMID: 36519888, PMID: 36215709, PMID: 36798414, PMID: 35217106). Discussion can be provided to compare the claim of the current manuscript to gain a broader view of whether it is a real case that currently available RAT kits are not sensitive enough to the Omicron variant.

Response:

 As suggested by the reviewer, we have added the discussion that comparing the currently available RATs that were reported are not affected by the emergence of the Omicron and the SARS-CoV-2 Rapid antigen test 2.0 and RAT 2.0 nasal. 

Pages: 18-19, Lines: 358-379.

“The performance of RATs for the detection of SARS-CoV-2, particularly the Omicron and its subvariants is influenced by several factors, including the viral genetics and the design of the test devices. Hence, each RAT is affected differently by viral mutation due to the difference in the inherent design of each RAT. Apart from the studies that reported the decline of the RAT for Omicron detection, other studies claimed that some of the currently available RATs are not affected by the emergence of Omicron, particularly when being used to assess the samples with high viral load. The DIAGNOVIR SARS-CoV-2 ultra-rapid antigen test, for example, was reported to demonstrate high performance in detecting most SARS-CoV-2 variants including Wuhan, Alpha (B1.17), Beta (B.1.351), Delta (B.1.617.2), and Omicron (B.1.1529). A study on several currently available RAT kits demonstrated a similar level of sensitivity to Omicron and non-Omicron variants (On/Go, 76.4% and 71.0%; iHealth, 73.0%, and 71.0%; QuickVue, 84.3%, and 74.3%; BinaxNOW, 69.7%, and 71.0%; and InBios, 66.3% and 64.5%, respectively). However, most of these studies only evaluated the sensitivity and specificity for the detection of Omicron B.1.1529 but not for the detection of currently circulating subvariants. A study reported that most of the Omicron subvariants accumulated amino acid substitutions in the N protein. Given that the N protein is an important target protein for most of the currently available RATs, any mutation in this protein may affect the sensitivity of RATs. Hence, the SARS-CoV-2 Rapid Antigen Test 2.0 and SARS-CoV-2 Rapid Antigen Test 2.0 Nasal, which enable rapid and accurate detection of SARS-CoV-2, particularly the currently circulating Omicron subvariants such as BA.4 and BA.5, are considered promising to assist in the control of the person-to-person or closed-community transmission of these subvariants.”

6. WGS sequencing result was not available from the manuscript although authors declare Yes for all data are fully available without restriction.

Response:

As suggested by the reviewer, we have added the WGS sequencing result in the supplementary figure. 

7. Line 48, HKUI can be converted to HKU-1.

Response:

We corrected it as recommended by the reviewer. 

Page 3, Line 47.

8. Line 80-81, reference citation is required.

Response:

We have added the citation as suggested by the reviewer.

Page 4, Lines 79-81. 

Reviewer #2: 

This is a timely manuscript.

1. it is not audience friendly to place the figure legends under Materials and Methods.

Response:

We placed Figure 1 in the materials and methods to give a better picture of our study design. However, according to the journal requirement, during the submission, the figure should be submitted in a separate file, while the figure legends have to be placed close to the figure citation (page 6, line 105), that’s why in the current version (submitted version) of our manuscript, only figure legend appears under the Materials and Methods. 

2. More details about the RT-PCR and WGS shall be provided, including commercial sources and Cat#.

Response:

We have added the catalog numbers for the RT-qPCR and the WGS system used in this study.

Pages: 7-8, lines: 144-146, 159 and 168

---

## [Decision Letter · Decision Letter 1]

4 Jul 2023

PONE-D-23-10862R1Efficacy of novel SARS-CoV-2 rapid antigen tests in the era of omicron outbreakPLOS ONE

Dear Dr. Kim,

Thank you for submitting your manuscript to PLOS ONE. After careful consideration, we feel that it has merit but does not fully meet PLOS ONE’s publication criteria as it currently stands. Therefore, we invite you to submit a revised version of the manuscript that addresses the points raised during the review process.

We look forward to receiving your revised manuscript.

Kind regards,

Ziwei Ye, Ph.D

Academic Editor

PLOS ONE

Journal Requirements:

Reviewers' comments:

Reviewer's Responses to Questions

**Comments to the Author**

1. If the authors have adequately addressed your comments raised in a previous round of review and you feel that this manuscript is now acceptable for publication, you may indicate that here to bypass the “Comments to the Author” section, enter your conflict of interest statement in the “Confidential to Editor” section, and submit your "Accept" recommendation.

Reviewer #1: (No Response)

2. Is the manuscript technically sound, and do the data support the conclusions?

Reviewer #1: Yes

3. Has the statistical analysis been performed appropriately and rigorously? 

Reviewer #1: Yes

4. Have the authors made all data underlying the findings in their manuscript fully available?

Reviewer #1: Yes

5. Is the manuscript presented in an intelligible fashion and written in standard English?

Reviewer #1: Yes

6. Review Comments to the Author

Reviewer #1: Reply to response 1:

The hyperlinks list in reference 24 and 25 might not be useful. Please revise the hyperlinks.

Reply to response 2:

Satisfied.

Reply to response 3:

The clarification is useful. Still, please provide reference supporting the use of variant calling or provide description to how exactly it works to identify variants from WGS data.

Reply to response 4:

Satisfied.

Reply to response 5:

Satisfied.

Reply to response 6:

WGS data is provided. However, please provide more description for how figure S1 could be co-referencing table 4. Value for sample2 4, 21, 23, 28 and 46 look low. Are they representing negative from the RAT tests? Possible to provide indication on figure S1 for RAT result?

Reply to response 7:

Satisfied.

Reply to response 8:

Satisfied.

7. PLOS authors have the option to publish the peer review history of their article (what does this mean?). If published, this will include your full peer review and any attached files.

Reviewer #1: **Yes: **Cheung Pak Hin Hinson

---

## [Author Response · Author response to Decision Letter 1]

24 Jul 2023

Journal Requirements:

Response: We have checked all the references and confirmed that none of them have been retracted as mentioned, hence a kind reminder of which reference became a concern will greatly help us to revise it. Other than reference numbers: 25,26,27 that were added to answer the reviewer comments, no other changes were made to the reference list. 

Reviewer response:

Reviewer #1: 

Reply to response 1:

The hyperlinks list in references 24 and 25 might not be useful. Please revise the hyperlinks.

Response: As suggested by the reviewer, we have revised the hyperlinks for mentioned references.

Reference 23 and 24. 

Reply to response 3:

The clarification is useful. Still, please provide a reference supporting the use of variant calling or describe how exactly it works to identify variants from WGS data.

Response:

The Omicron subvariants were determined through “variant calling” in the WGS analysis, which is the process by which the variants were identified according to the sequence data deposited in the database (GISAID). Briefly, the quality of the sequencing data in the format of Fastq was checked using the FastQC (v0.10.1) program, meanwhile, the Base Quality, Duplication level, and GC content were visually checked. Subsequently, the trimming procedure was employed to filter the reads to remove the low-quality sequences. The sickle (v.1.3.3) program was used for the entire filtering process. Further, the reads were aligned to the SARS-CoV-2 reference using BWA (B0.7.17) program, and the duplication reads generated through PCR were removed using GATK (v.4.0.2.1) program. Finally, the stand_call_conf - 30.0, stand_emit_conf -10.0, and dcov - 1,000 options were applied to find mutations within the sequence which already clear out from any duplication, and the variant analysis was performed using GATK (v.4.0.2.1). Further, the annotation of mutation from GATK (v.4.0.2.1) was finalized using the SnpEff (V.4.1) program and the SARS-CoV-2 strain of each sample was determined after comparing it with the sequences database in the global initiative for sharing all influenza data (GISAID, https://gisaid.org/) and with the mutation databases that were obtained from the lineage-specific mutation information database provided by the outbreak.info, CoVariants, and COVID CG.

Pages: 8-9, Lines: 172-189

Reply to response 6:

WGS data is provided. However, please provide more description for how Figure S1 could be co-referencing Table 4. Value for sample2 4, 21, 23, 28, and 46 look low. Are they representing negatively from the RAT tests? Possible to indicate Figure S1 for the RAT result?

Response:

Figure S1 represents the result of WGS hence do not directly represent the RAT test results. We have corrected the position where Figure S1 was being cited, so it won’t cause ambiguity or misunderstanding. 

Additionally, among all samples assessed in this study, three were confirmed negative by RATs but positive by omicron PCR, one was confirmed negative by both RAT and Omicron PCR, and 52 samples were confirmed positive by both RAT and Omicron PCR. Therefore, for the WGS analysis, we assessed 55 samples, comprised of 52 samples positive by both RAT and Omicron PCR, and three samples confirmed positive by only Omicron PCR. 

Among the samples 4,21,23,28, and 46 that have low values, only sample 4 was confirmed negative by RATs, hence the low values in WGS do not directly relate to the negativity of RAT. However, all these samples have low values which resulted in uncertainty of the lineage where the samples belong. 

We also added some descriptions related to this issue in the WGS result section. 

 “We also compared the result from the RATs assay with that from WGS. In this study, we assessed 55 specimens for WGS, comprised of 52 confirmed positive by both RATs and Omicron PCR, and three confirmed positive by only Omicron PCR. One specimen was excluded from WGS analysis due to the negative result from both RATs and Omicron PCR tests.”

Page 14, Lines: 265-269

“The six specimens identified as “uncertain” showed a low value of the variables that were being assessed in the WGS analysis, hence making them difficult to be distinguished into specific lineages (S1 Fig). “

Page 14, Lines: 277-279

---

## [Editor Report · Decision Letter 2]

31 Jul 2023

Efficacy of novel SARS-CoV-2 rapid antigen tests in the era of omicron outbreak

PONE-D-23-10862R2

Dear Dr. Kim,

We’re pleased to inform you that your manuscript has been judged scientifically suitable for publication and will be formally accepted for publication once it meets all outstanding technical requirements.

Kind regards,

Ziwei Ye, Ph.D

Academic Editor

PLOS ONE

---

## [Editor Report · Acceptance letter]

2 Aug 2023

PONE-D-23-10862R2 

Efficacy of novel SARS-CoV-2 rapid antigen tests in the era of omicron outbreak 

Dear Dr. Kim:

I'm pleased to inform you that your manuscript has been deemed suitable for publication in PLOS ONE. Congratulations! Your manuscript is now with our production department. 

Kind regards, 

on behalf of

Dr. Ziwei Ye 

Academic Editor

PLOS ONE